# Global circRNA expression changes predate clinical and histological improvements of psoriasis patients upon secukinumab treatment

Sabine Seeler[1], Liviu-Ionut Moldovan[1], Trine Bertelsen[2], Henrik Hager[3], Lars Iversen[2], Claus Johansen[2], Jørgen Kjems[4,5], Lasse Sommer Kristensen[1] *

1 Faculty of Health, Department of Biomedicine, Aarhus University, Aarhus, Denmark, 2 Department of Dermatology, Aarhus University Hospital, Aarhus, Denmark, 3 Department of Clinical Pathology, Vejle Hospital, Vejle, Denmark, 4 Interdisciplinary Nanoscience Center (iNANO), Aarhus University, Aarhus, Denmark, 5 Department of Molecular Biology and Genetics (MBG), Aarhus University, Aarhus, Denmark

☯ These authors contributed equally to this work.
* lasse@biomed.au.dk

**Data Availability Statement:** All relevant data are within the paper and its Supporting Information files.

## Abstract

Psoriasis is a common chronic inflammatory skin disease accompanied by heterogenous clinical and histological features, including a characteristic keratinocyte hyperproliferation and dermal immunogenic profile. In addition, psoriasis is associated with widespread transcriptomic alterations including changes in microRNA (miRNA) and circular RNA (circRNA) abundance, which constitute non-coding RNA (ncRNA) classes with specific regulatory capacities in diverse physiological and pathological processes. However, the knowledge about the expression dynamics of ncRNA during psoriasis treatment is sparse. To elucidate the dynamics of miRNA and circRNA abundance during secukinumab (anti-IL-17A) treatment, we studied their expression patterns in skin biopsies from 14 patients with severe plaque-type psoriasis before and during an 84-day secukinumab therapy at day 0, 4, 14, 42, and 84 using NanoString nCounter technology. We found a comprehensive downregulation of the majority of investigated circRNAs and specific alterations in the miRNA profile, including an upregulation of miR-203a-3p, miR-93-5p, and miR-378i in lesional compared to non-lesional skin before treatment. During treatment, the circRNAs progressively returned to the expression levels observed in non-lesional skin and already four days after treatment initiation most circRNAs were significantly upregulated. In comparison, for miRNAs, the normalization to baseline during treatment was delayed and limited to a subset of miRNAs. Moreover, we observed a strong correlation between multiple circRNAs, including ciRS-7 and circPTPRA, and the psoriasis area and severity index (PASI). Similar pronounced correlations could, however, not be found for miRNAs. Finally, we did not observe any significant changes in circRNA expression in peripheral blood mononuclear cells during treatment. In conclusion, we uncovered a rapid shift in global circRNA abundance upon anti-IL-17A treatment, which predated clinical and histological improvements, and a strong correlation with PASI, indicating a biomarker potential of individual circRNAs.

**Funding:** This project was supported by the European Union's Horizon 2020 research and innovation programme under the Marie Skłodowska-Curie [grant no. 721890; J.K.; https://marie-sklodowska-curie-actions.ec.europa.eu/], the Leo Foundation [grant no. LF-OC-20-000376; L.S.K.; https://leo-foundation.org/en/], Villum Foundation [grant no. 00013393; J.K.; https://veluxfoundations.dk/en], and Lundbeck foundation [grant no. R307-2018-3433; L.S.K.; https://lundbeckfonden.com/en]. The funders had no role in study design, data collection and analysis, decision to publish, or preparation of the manuscript.

**Competing interests:** I have read the journal's policy and the authors of this manuscript have the following competing interests: L.I. served as a consultant and/or paid speaker for and/or participated in clinical trials sponsored by AbbVie, Almirall, Amgen, AstraZeneca, BMS, Boehringer Ingelheim, Celgene, Centocor, Eli Lilly, Janssen Cilag, Kyowa, Leo Pharma, MSD, Novartis, Pfizer, Samsung, and UCB. This does not alter our adherence to PLOS ONE policies on sharing data and materials.

## Introduction

Psoriasis is a systemic chronic inflammatory skin disease characterized by erythematous and scaly plaques, a pronounced dermal infiltration of immune cells, and epidermal hyperplasia due to keratinocyte hyperproliferation [1–4]. The psoriasis pathogenesis is complex, with genetic susceptibility and environmental triggers influencing the initial aberrant immune response in the skin, leading to a sustained skin inflammation that, together with various comorbidities, significantly affect the patients' quality of life [1–6]. During the initial immune response, activated plasmacytoid dendritic cells (DCs) induce the maturation of proinflammatory DC populations via type I interferon (IFN) secretion, which in turn stimulates the differentiation of T-helper ($T_H$) cells, including $T_H1$, $T_H17$, and $T_H22$ via interleukin (IL)-12 and IL-23 [7,8]. These activated CD4$^+$ and CD8$^+$ T cells belong to the inflammatory cells found in psoriatic skin lesions and are influencing disease progression, specifically keratinocyte hyperproliferation, through the production of proinflammatory cytokines, such as IL-17A/F, tumor necrosis factor alpha (TNFα), IFNγ, and IL-22 [5,8,9]. Thus, the IL-23/IL-17 inflammatory axis plays a crucial role in psoriasis pathogenesis and is, hence, target of several targeted psoriasis treatments.

IL-17A can be selectively inhibited by secukinumab, a fully human monoclonal IgG1/κ antibody (mAb), which has been approved to treat moderate-to-severe plaque psoriasis, psoriatic arthritis, as well as ankylosing and axial spondyloarthritis [10,11]. Recent reports have shown that as early as one week after initiation of secukinumab treatment, major changes occurred in psoriasis-related transcripts, along with improved clinical scores and histologic psoriasis features at week 12 [12]. In addition, we showed previously that already on day four of treatment, 80 genes were differentially expressed, one of them being *NFKBIZ* [13], which encodes for a nuclear inhibitor of NF-κB, shown to be an important regulator of psoriasis progression [14].

Non-coding RNAs (ncRNAs), including microRNAs (miRNAs) and circular RNAs (circRNAs), are also known to be deregulated in psoriasis [15,16], but the impact of treatment, and particularly of secukinumab, on these transcripts has not been uncovered thus far. While the functional roles of miRNAs as post-transcriptional regulators of gene expression are well established [17], the potential functions of circRNAs are still under active investigation. While many lowly expressed circRNAs are likely to be mere by-products of aberrant RNA splicing events [18–20], many circRNAs may, indeed, function through a multitude of different mechanisms [21–26]. Here, it is evident that several gene loci produce functional circRNAs with gene-regulatory potential through interaction with miRNAs [21,22,27] and/or proteins [23,24,28]. In addition, a cell- and tissue-specific circRNA expression has been uncovered by transcriptomic studies based on high-throughput RNA sequencing and specific bioinformatic algorithms [29,30]. Lastly, multiple studies indicate that circRNAs could play important roles in a wide range of physiological and disease processes, including neurogenesis [31–34], atherosclerosis [35], chronic inflammatory diseases [36,37], and cancer [38,39].

In this study, we investigated changes in the circRNA and miRNA expression in skin biopsies and circRNA expression in peripheral blood mononuclear cells (PBMCs) during 84 days of subcutaneous secukinumab administration in a previously characterized psoriasis patient cohort [13].

## Materials and methods

### Patient cohort and sample preparation

For this study 14 patients with severe plaque-type psoriasis were treated with 300 mg of secukinumab subcutaneous injections at weeks zero, one, two, and four and then every four weeks,

as described previously [13]. Blood samples and four-millimeter lesional and non-lesional punch biopsies were obtained prior to treatment at day zero, as well as blood and lesional biopsies on days 4, 14, 42, and 84 (L4, L14, L42, and L84) after commencing treatment. Biopsies were taken as paired samples from the same body region, and the non-lesional biopsies were taken at a minimum distance of five centimeters from a lesional plaque. At each study visit, patients were evaluated clinically, and clinical improvement was assessed using changes in psoriasis area and severity index (PASI). In addition, skin biopsies of eight healthy age- and gender-matched (average age = 38.38 years; n[male controls] = 5, n[female controls] = 3) controls were compared to non-lesional/lesional samples from 8 out of the 14 psoriasis patients (average age = 42.38 years; n[male controls] = 6, n[female controls] = 2). The biopsies were snap-frozen in liquid nitrogen and stored at -80˚C until further use. For RNA extraction, psoriatic patients' punch biopsies were transferred to 1 ml of -80˚C cold RNAlater-ICE (Ambion inc., Austin, TX, USA). The study was carried out following the Declaration of Helsinki and a written and signed informed consent was obtained from each patient. Only adult patients above the age of 18 were enrolled. The study was approved by The Central Denmark Region Committees on Health Research Ethics (M-20090102).

## RNA extraction from biopsies

RNA extraction was performed as described previously [16]. Briefly, samples were stored at -80˚C and moved to -20˚C for 24 hours before RNA purification. For RNA purification, biopsies were removed from RNAlater-ICE, 175 µl of SV RNA Lysis Buffer including β-mercaptoethanol was added (SV Total RNA Isolation System; Promega, Madison, WI, USA), and samples were homogenized. RNA purification including DNase treatment was completed according to the manufacturer's instructions (SV Total RNA Isolation System; Promega, Madison, WI. USA).

## RNA chromogenic in situ hybridization (CISH)

CISH on ciRS-7 was performed on formalin-fixed and paraffin-embedded skin punch biopsy sections using an adapted protocol of the RNAScope 2.5 high-definition procedure (Advanced Cell Diagnostics [ACD], Hayward, CA, USA), as previously described [16,40]. Briefly, after pretreatment of paraffin sections, they were hybridized overnight with 12 ZZ-pairs (Probe-Hs-CDR1-AS-No-XMm, 510,711, ACD) targeting ciRS-7. The ZZ-pairs were amplified using seven amplification steps, including a tyramide signal amplification step (TSA-DIG; NEL748001KT, PerkinElmer, Skovlunde, Denmark) and an alkaline phosphatase-conjugated sheep anti-DIG FAB fragment (Roche, Basel, Switzerland). Lastly, visualization was facilitated with liquid permanent red (DAKO, Glostrup, Denmark) and Mayer's hematoxylin counterstaining.

## CircRNA expression analyses using NanoString nCounter technology

We used a custom designed NanoString CodeSet of capture- and reporter probes targeting 100 bp of the backsplicing junctions (BSJs) of the 50 most abundant circRNAs in lesional and non-lesional skin biopsies of psoriasis patients [16] (S1 File). For one out of the 50 most abundant circRNAs (circMAN1A2 isoform), we were not able to find a probe design that distinguishes between isoforms. The circRNAs termed circVPS16 and CDR1 in the NanoString panel are referred to as circPTPRA and ciRS–7 in this study, respectively. Additionally, twelve linear genes were included, seven of which were stable reference genes, as reported earlier [41], and five additional mRNAs that were selected due to their coding potential for RNA-binding proteins (RBPs), reported to influence circRNA expression (*FUS*, *ADAR*, *HNRNPL*, *QKI*, and

*DHX9*) [16]. For the experiments characterizing circRNA profiles in PBMCs and healthy control skin, seven additional circRNAs were included (S1 File). An input of 70 ng (Data presented in S3 File) or 100 ng (Data presented in S2 and S4 Files) of total RNA was used for 20 hours of hybridization and the subsequent nCounter™ *SPRINT* (NanoString Technologies, Seattle, WA, USA) run. The raw data was normalized using the nSOLVER 3.0 software (NanoString Technologies); first, a positive control normalization was performed using the geometric mean of all but the last positive controls. Then a second normalization with the geometric mean of the three most stable linear reference genes (*RPL0*, *RPL19*, and *ACTB*, or *RPL0*, *RPL19*, and *GUSB*) was performed.

## Isolation of PBMCs

Fresh whole blood was taken from psoriatic patients and added to EDTA containing tubes. The EDTA blood was mixed 1:1 with Hanks buffered saline and separated by centrifugation using Lymphoprep (Axis-Shield, Cambridgeshire, UK). The PBMC layer was subsequently isolated and washed with Hanks buffered saline. The centrifuged pellet constituted the PBMCs.

## miRNA expression analysis using NanoString nCounter technology

For miRNA expression analysis the NanoString human v3 miRNA panel (NanoString Technologies), targeting 799 miRNAs, was used according to the manufacturer's instructions. Briefly, 100 ng of total RNA was used with 20 hours hybridization. Subsequently, the hybridization reaction was analyzed using the nCounter™ *SPRINT* (NanoString Technologies) platform and the raw data were processed using the nSOLVER 3.0 software; first, a background subtraction of the geometric mean of the negative controls and a positive control normalization using the geometric mean of all positive controls was performed. A second normalization followed using the geometric mean of the housekeeping genes *RPL19* and *RPL0*. In addition, a sample calibration was performed for batch effect correction introduced by using code sets with different LOT numbers.

## Statistical analysis

All statistical tests were performed using Prism 9 (GraphPad, La Jolla, CA, USA). Volcano plots for changes in circRNA, miRNA, and differences in expression levels of *ADAR*, *QKI*, *FUS*, *DHX9*, and *HNRNPL* were generated by multiple paired t-test with correction for multiple comparisons (FDR-Benjamini-Hochberg). For comparison of lesional or non-lesional to healthy control skin, we performed multiple unpaired t-tests with correction for multiple comparisons (FDR-Benjamini-Hochberg). Linear regression was used to assess the potential correlation between log2-transformed normalized counts and the PASI by checking whether the slope was significantly non-zero. All p-values were two-tailed and considered significant if $p < 0.05$.

## Results

### Normalization of circRNA expression profiles in psoriatic skin upon secukinumab treatment

To explore the impact of secukinumab treatment on circRNA expression profiles in the skin of patients with severe plaque-type psoriasis, we used a custom-designed NanoString panel targeting the top 50 most abundant circRNAs in skin of psoriasis patients [16]. In addition, the panel contained five mRNAs selected based on their coding potential for RBPs with influence on circRNA biogenesis (*FUS* [42], *ADAR* [43], *HNRNPL* [44], *QKI* [45], and *DHX9* [20]).

First, to determine the overall effects on circRNA expression patterns of secukinumab during 84 days of treatment, we employed principal component analysis (PCA) (Fig 1A, S2 File). Here, we observed a grouping of samples from non-lesional with lesional skin at day 42 and 84 of treatment. Accordingly, samples from untreated lesional skin showed a similar expression profile as lesional skin at treatment day 4 and 14. Considering the mean circRNA expression across patients, lesional and non-lesional skin samples were markedly different from each other at day zero, before treatment and a clear progressive effect of the treatment was observed with time, showing that the patients' lesional skin gradually returned towards the circRNA expression pattern of non-lesional skin (Fig 1B). Strikingly, the differences in mean circRNA expression at day 84 in comparison to lesional skin at baseline seemed to be even more drastic relative to non-lesional skin, and thus appeared to surpass the non-lesional skin as reference point. Thus, we also investigated if circRNA patterns showed variation between non-lesional and healthy control skin but did not find significant differences (S1A and S1B Fig, S3 File).

To illustrate the behavior of individual circRNAs throughout the treatment period, we also plotted the average circRNA expression for non-lesional and lesional skin during treatment as a heatmap with unsupervised hierarchical clustering of the circRNAs (Fig 1C), as well as patient-specific expression patterns (S1C Fig). Consistent with the PCA plot, a striking restoration of the expression patterns was observed for most circRNAs, as treatment progressed. Here, we found that the normalization started already at the first investigated time point; only four days after treatment commenced. The expression of most circRNAs were upregulated during treatment, however, some circRNAs clustered separately from the majority due to distinct expression profiles. For example, a profound downregulation was observed for circZKSCAN1, circCAMSAP1, circDDX21, circERC1, circFBXW7, circHIPK3, and circGDI2 at day 84 relative to day 0.

In line with our previous reports [16,46], we observed a considerable circRNA downregulation in lesional relative to non-lesional skin before treatment started (Fig 1D), as well as relative to healthy control skin (S1D Fig). In lesional relative to non-lesional skin among the 43 significantly differentially expressed circRNAs (adjusted p-value < 0.05), 39 were downregulated, 25 and 10 of them more than 1.5- and 2-fold, respectively. Upon day 4 and 14 of secukinumab treatment, all investigated circRNAs were upregulated relative to untreated lesional skin (Fig 1E and 1F, respectively). Intriguingly, we found 88% (22 out of 25) of the more than 1.5-fold deregulated circRNAs at baseline to be significantly reversed towards a non-lesional expression profile at day four. However, only circZC3H6 increased more than 1.5-fold in this time frame.

Next, we examined which of the initially downregulated circRNAs (Fig 2A, left) showed a pronounced inverted expression pattern at 42 and 84 days after treatment commenced. At day 42, circTULP4, circPARD3, circPTPRA, and ciRS-7 were amongst the most significantly upregulated circRNAs and two circRNAs were significantly downregulated (circCAMSAP1, circERC1) (Fig 2A, middle). At day 84, among the 43 differentially expressed circRNAs, 38 were upregulated, 10 of them more than 2-fold, and ciRS-7 more than 4-fold relative to lesional skin before treatment (Fig 2A, right), making it also the most upregulated circRNA after 84 days of treatment. Interestingly, only six circRNAs were differentially expressed at day 84 relative to non-lesional skin prior to treatment (S1E Fig).

Together, these analyses demonstrated a progressive normalization of circRNA expression profiles during secukinumab treatment of psoriasis patients. Additionally, this showed that many of the most downregulated circRNAs in lesional skin relative to non-lesional skin prior to treatment became significantly upregulated already four days after treatment was initiated.

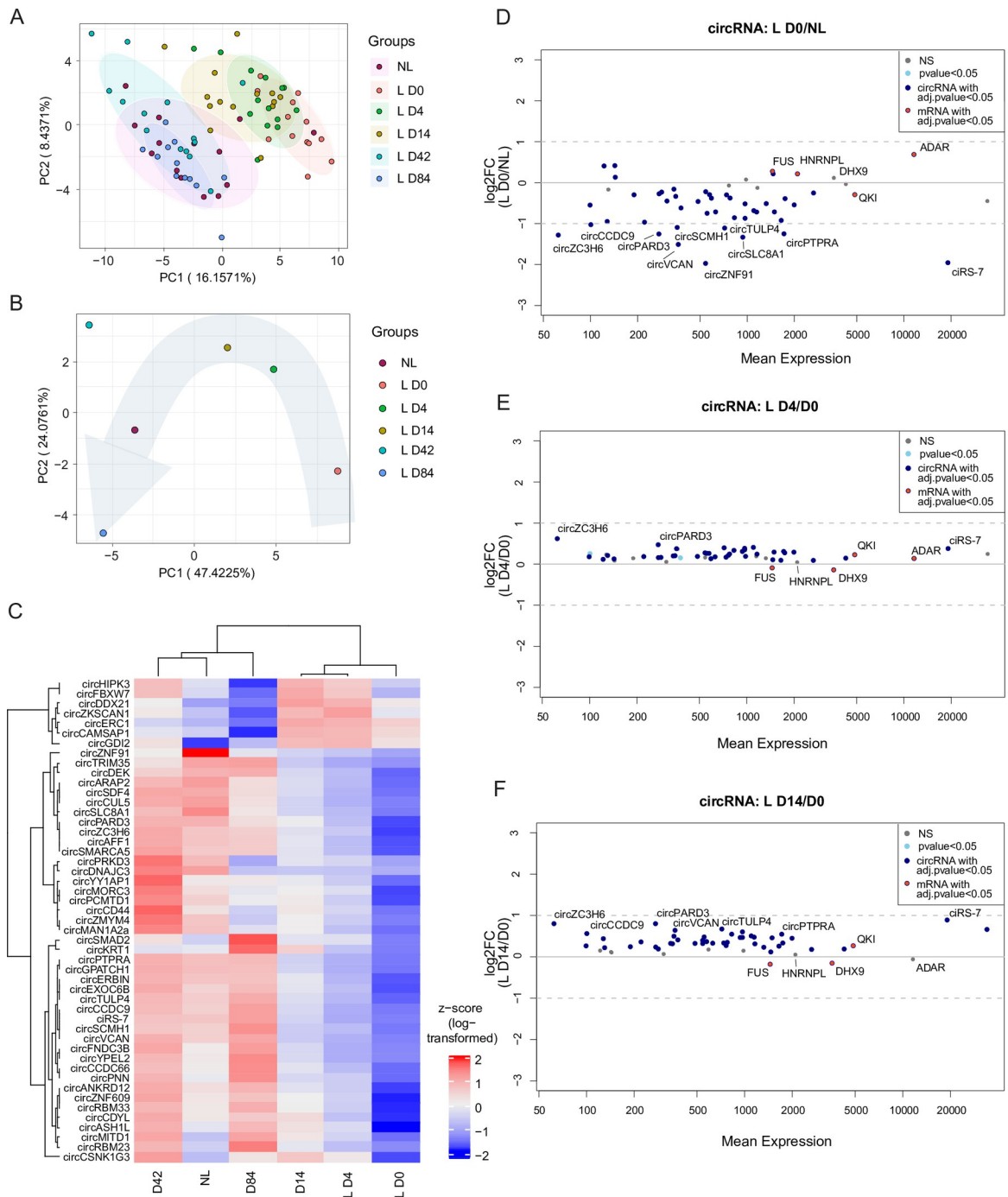

**Fig 1. Alterations in global circRNA expression in non-lesional (NL) and lesional (L) skin of psoriasis patients before and during secukinumab treatment.** (A-B) Principal component analysis based on circRNA expression levels in individual patients (A) and mean circRNA expression levels between patients (B) from non-lesional and paired lesional psoriasis skin before (L D0) and after 4 (L D4), 14 (L D14), 42 (L D42), and 84 (L D84) days of treatment. (C) Heatmap with unsupervised hierarchical clustering of mean circRNA expression (as z-score of log-transformed values) between patients from non-lesional and paired lesional psoriasis skin dependent on the day of treatment. (D-F) MA plots depicting changes in circRNA and mRNA levels between lesional D0 skin in contrast to non-lesional skin (D), as well as D4 (E) and D14 (F) in contrast to D0 lesional skin. Depicted are the normalized mean expression relative to the log2 fold change (log2FC); n(NL, L D4, L D14, and L D43) = 14, n(L D0) = 13, and n(L D84) = 12. All mRNAs are labelled (*FUS*, *HNRNPL*, *DHX9*, *ADAR*, and *QKI*).

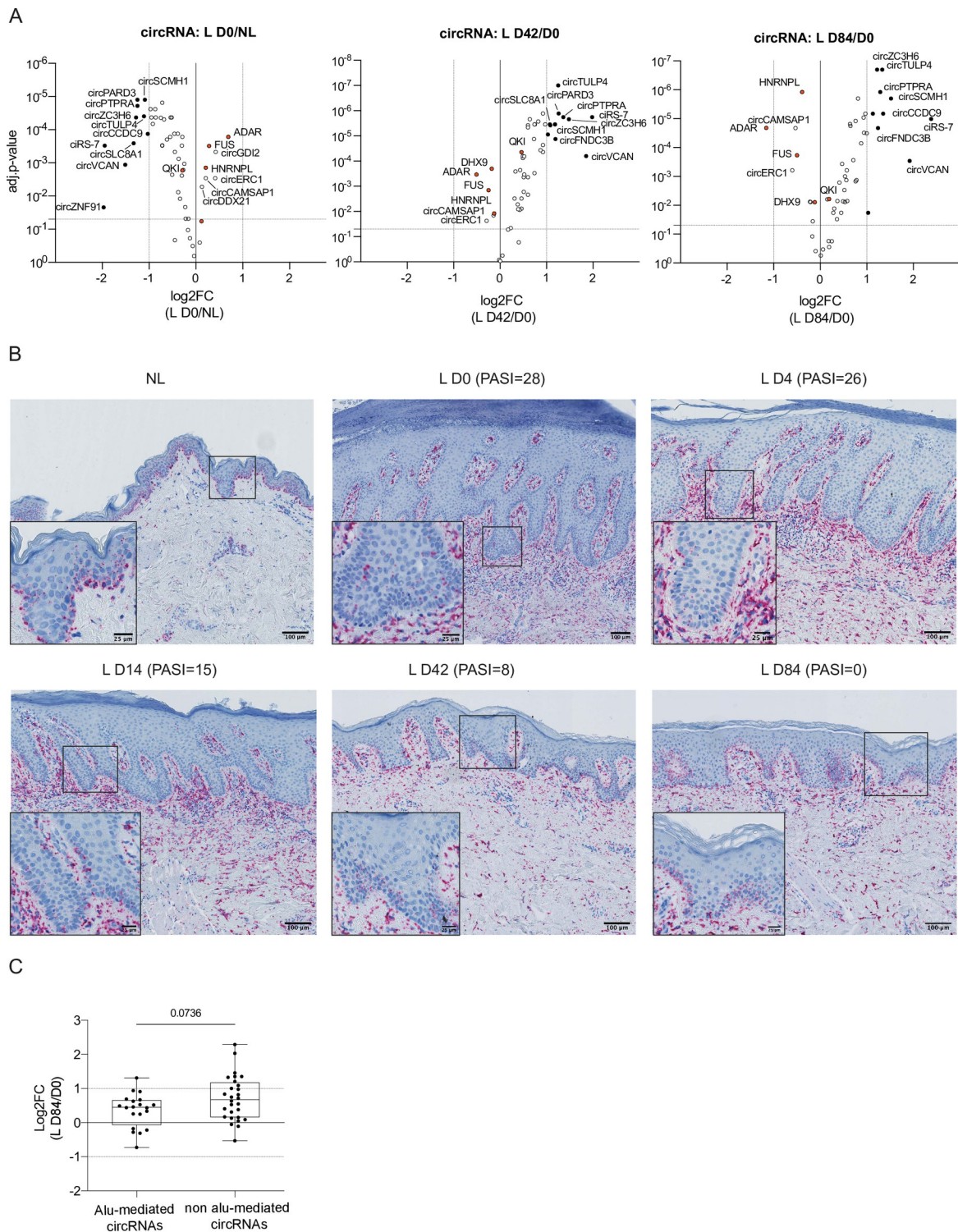

**Fig 2. Changes in circRNA expression and ciRS-7 localization in non-lesional (NL) and lesional (L) skin of psoriasis patients before and during secukinumab treatment.** (A) Volcano plots show changes in specific circRNA and mRNA expression in lesional psoriasis skin before treatment (L D0; left) and during secukinumab treatment on day 42 (middle) and 84 (right). Plots depict adjusted (adj.) p-values relative to log2FC. Multiple paired t-test with correction for multiple comparisons (FDR-Benjamini-Hochberg); n(NL, L D4, L D14, and L D43) = 14, n(L D0) = 13, and n(L D84) = 12. Black points depict circRNAs with a |log2FC| > 1 and adj.-p-value < 0.05. Orange points depict mRNAs (*FUS*, *HNRNPL*, *DHX9*, *ADAR*, and *QKI*). (B) Hematoxylin staining combined with RNA CISH for ciRS-7 during treatment with secukinumab. Paired biopsies from non-lesional and lesional skin before (L D0), and 4, 14, 42, and 84 days after

treatment initiation. Four micrometer sections of paraffin-embedded biopsies with one representative patient shown. Scale bar: 100 μm and 25 μm in zoom-in. (C) Boxplot showing the median log2FC of individual circRNAs in lesional D84 relative to D0 skin dependent on whether or not they characterize as Alu-mediated circRNA (presence of IAEs within 2300 nucleotide regions flanking the BSJs). Twenty circRNAs out of 48 circRNAs were considered Alu-mediated. Depicted are median values with whiskers extending to min. and max. values. Two-tailed Mann-Whitney test; n = 11. BSJ = backsplicing junction; CISH = chromogenic *in situ* hybridization; IAE = Inverted Alu elements.

### ciRS-7 exhibits particular histological expression patterns in the post-treatment psoriatic skin

Because ciRS-7 stood out as one of the most significantly upregulated circRNAs during treatment, we visualized the spatial expression pattern of ciRS-7 on representative skin samples from psoriasis patients undergoing secukinumab treatment using an RNA CISH assay. While we observed ciRS-7 in non-lesional skin to be predominately expressed in the epidermis, especially within the basal layer, a marked downregulation of ciRS-7 in the epidermis of lesional skin was found (Fig 2B), in line with previous observations [16]. On the contrary, we found a strong ciRS-7 presence in the dermal compartments of lesional skin, particularly before and early during treatment. Thus, the observed ciRS-7 spatial expression patterns during treatment confirmed a progressive restoration of ciRS-7 levels in the epidermis over 84 days of treatment, further supporting our NanoString experiments.

### Alu-mediated circRNAs were not significantly deregulated upon secukinumab treatment

In lesional relative to non-lesional skin before treatment initiation (Fig 2A, left; S2 File), *FUS*, *ADAR*, and *HNRNPL* were significantly upregulated and *QKI* significantly downregulated, however, none of these mRNA transcripts showed changes in expression greater than 2-fold. Strikingly, the only difference in the considered mRNA transcripts during treatment was the downregulation of *ADAR* at day 84 relative to lesional skin at baseline (Fig 2A, right). ADAR is a suppressor of the biogenesis of a subset of circRNAs due to its RNA editing capacity interfering with base pairing between inverted Alu elements (IAEs) flanking the circRNA-containing exon(s) [43]. Therefore, we investigated changes in expression levels of circRNAs that are predicted to be formed through base pairing of IAEs. To this end, we categorized our circRNAs of interest into either "Alu-mediated circRNAs", if the circularized exons are flanked by IAEs within a 2300 nucleotide range relative to the BSJ, or "non Alu-mediated circRNAs" [16]. Next, we compared the relative circRNA changes at day 84 of treatment relative to baseline, as *ADAR* was significantly downregulated at this time. However, we did not find Alu-mediated circRNAs to be more drastically upregulated than the remaining circRNAs (Fig 2C). Thus, the increase in circRNA expression is most likely not a consequence of less ADAR modification within IAE flanking the circRNAs.

### Secukinumab did not significantly alter circRNA expression patterns in PBMCs during treatment

Next, to explore whether the secukinumab-induced circRNA expression changes are limited to the skin or whether they are of systemic nature, we analyzed RNA extracted from PBMCs from the same patient cohort at the same time points, using our custom circRNA NanoString panel. Initially, PCA and unsupervised hierarchical clustering were used for the evaluation of synchronized changes between the circular transcripts of interest during secukinumab treatment in PBMCs (S2A and S2B Fig, S4 File). Here, we did not identify a clear clustering of samples based on the treatment time. Similarly, we did not observe any differentially expressed

circRNAs at any time point in the PMBCs relative to their levels at day zero of treatment (S2C–S2F Fig). When considering absolute counts for two of the most deregulated circRNAs in psoriatic skin, ciRS-7 and circPTPRA, we did not see any significant changes over the course of treatment either (S2G and S2H Fig, respectively). These results imply that neither secukinumab nor the degree of psoriasis have an impact on the circRNA expression in PBMCs.

## Changes in miRNA expression profile in psoriatic skin during secukinumab treatment

With previous publications showing a putative interaction network between circRNAs and miRNAs [21,22,27], as well as a functional role of miRNAs in psoriasis progression [15,47,48], we assessed the miRNA expression landscape in the same psoriasis patient cohort at the same time points. To this end, we used a pre-designed NanoString miRNA panel covering 799 commonly found miRNAs, of which 162 exceeded our average expression cut-off of 20 counts across all samples. First, we investigated miRNA expression changes for the 162 miRNAs by PCA, either considering individual patients (Fig 3A, S5 File) or an average expression across patients (S3A Fig), and unsupervised hierarchical clustering of miRNAs (Fig 3B). Here, we found a tendency of lesional samples at baseline to show a similar expression profile as samples upon 4 and 14 days of treatment and accordingly day 42 and 84 samples seemed to group with non-lesional skin (Fig 3B). In addition, we again checked for miRNA expression changes between non-lesional and healthy control skin but found no significant differences (S3B and S3C Fig, S6 File). In contrast to the changes in circRNA expression profiles, the differences in miRNA levels upon treatment initiation appeared overall delayed (Fig 3B–3D), less pronounced (Fig 3A), and induced miRNA expression changes that were independent of the psoriasis-related miRNA alterations between lesional and non-lesional skin (S3D and S3E Fig). In particular, we observed no skewed deregulation in miRNA levels in lesional compared to non-lesional (Fig 3C, S5 File), but in healthy control skin (S3F Fig, S6 File). However, we found a group of significantly upregulated miRNAs (Fig 3C), including miR-203a-3p, which was previously shown to be increased in psoriatic lesions [15,47,48], and miR-93-5p, which was described to be implicated in keratinocyte proliferation [49]. In addition, we observed an upregulation of miR-223-3p, which was proposed previously as biomarker in PBMCs from psoriasis patients [50]. During secukinumab treatment, a subset of the deregulated miRNAs at baseline were reversed in their expression (Fig 3F; colored dots).

As described above, we observed a partial normalization already after four days of treatment for 88% (22 out of 25) of the more than 1.5-fold deregulated circRNAs at baseline. For the miRNAs changed at baseline, we found only 3.2% (2 out of 62) to be reversed significantly at day four and most miRNAs were still upregulated (Fig 3D). In contrast, after 14 days we observed a normalization for a subset of miRNAs and after 42 days, lesional skin samples showed a universal increase of miRNA levels, with seven miRNAs being significantly upregulated (Fig 3F; middle). Amongst the six significantly downregulated miRNAs were miR-203a-3p and miR-93-5p. At day 84 of secukinumab treatment, the global increase continued, with a total of 86 significantly upregulated miRNAs (Fig 3F; right). However, only three of these were reversed in their expression when considering miRNAs that were more than 2-fold changed at baseline. In contrast, 24 significantly downregulated miRNAs were found at day 84. Thirty-two miRNAs showed a reverse expression compared to baseline ($|log2FC|>0.5$ and p-value$<0.05$), while six miRNAs additionally seemed to be reversed in expression at day 42, including miR-203a-3p, miR-93-5p, miR-15a-5p, miR-378i, and miR-223-3p. Like the observations in circRNA analysis during secukinumab treatment, the miRNA levels after 42 days of

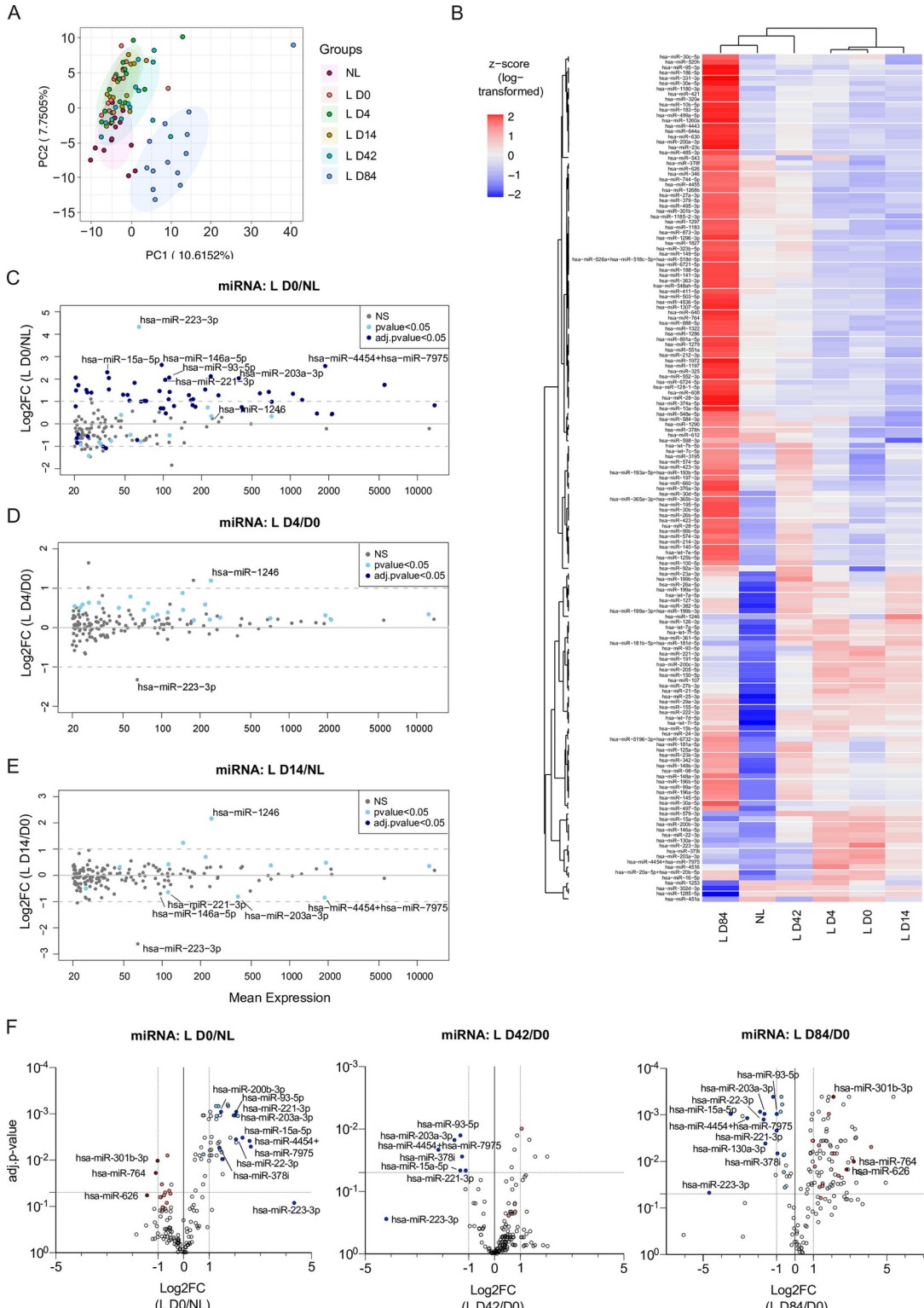

**Fig 3. Changes in miRNA expression profile in psoriatic skin during secukinumab treatment.** (A) Principal component analysis based on miRNA expression levels in individual patients from non-lesional and paired lesional psoriasis skin before (L D0) and after 4 (L D4), 14 (L D14), 42 (L D42), and 84 (L D84) days of treatment. (B) Heatmap with unsupervised hierarchical clustering of mean miRNA expression (as z-score of log-transformed values) between patients from non-lesional and paired lesional psoriasis skin dependent on the day of treatment. (C-E) MA plots depicting changes in miRNA levels between lesional

and non-lesional skin (C), lesional skin before treatment (L D0), and lesional skin during secukinumab treatment on day 4 (D) or 14 (E). Depicted are the log2FC relative to mean expression. Multiple paired t-test with correction for multiple comparisons (FDR-Benjamini-Hochberg). (F) Volcano plots depicting changes in miRNA levels between lesional and non-lesional skin (left), lesional skin before treatment (L D0) and lesional skin during secukinumab treatment on day 4 (middle) or 14 (right). All colored dots depict miRNAs that were deregulated in lesional skin before treatment and were reversed in their expression pattern after 42 and/or 84 days of treatment (|log2FC|>0.5 and p-value<0.05). Blue dots (dark and light blue) depict miRNAs with higher expression in lesional skin before treatment. Red dots (dark and light red) depict miRNAs with lower expression in lesional skin before treatment. Darker and lighter colors indicate miRNAs with |log2FC|>1 and |log2FC|>0.5 (in both D0 and D84), respectively; n(NL, L D4, L D14, and L D43) = 14, n(L D0 and L D84) = 13. Depicted are only miRNAs with an average expression above 20 counts (n = 162).

treatment (S3D Fig) seemed to surpass the miRNA levels of non-lesional skin at baseline and showed an additional upregulation after 84 days (S3E Fig).

In summary, like the normalization of the circRNA profile upon secukinumab treatment, we found a subset of the investigated miRNAs that returned to the expression profile of non-lesional skin upon treatment. However, in contrast to the circRNAs, the miRNA landscape showed a delayed normalization and significant alterations at 14 and 42 days of treatment, respectively. In addition, we found miR-203a-3p, miR-93-5p, miR-378i, and miR-223 expression to be upregulated in psoriatic lesions and to be reversed during day 42 to 84 of secukinumab treatment.

## CircRNA and miRNA expression in skin is significantly correlated with post-treatment changes in the PASI

The fact that miRNA and especially circRNA expression exhibited such a coordinated response during secukinumab treatment, prompted us to investigate their correlation with the PASI over time. Interestingly, all circRNAs, which were more than 2-fold downregulated in lesional relative to non-lesional skin before treatment, showed a significant negative correlation with the PASI over the course of treatment (Table 1). CircPTPRA, ciRS-7, and circTULP4 displayed a strong negative correlation with PASI (r = -0.77, -0.73, and -0.67, respectively) (Table 1, Figs 4A, 4B and S4A). Interestingly, a positive correlation was found between the significantly upregulated circRNAs at baseline and changes in the PASI, albeit less pronounced (S7 File).

Together these analyses showed that the circRNA expression changes during secukinumab treatment are either negatively or positively correlated with the PASI depending on whether

**Table 1. Correlation between circRNA expression changes and PASI during 84 days of secukinumab treatment.**

| circRNA | R | R squared | p-value |
|---|---|---|---|
| circPTPRA | - 0.7681 | 0.5900 | <0.0001 |
| ciRS-7 | - 0.7284 | 0.5305 | <0.0001 |
| circSCMH1 | - 0.7068 | 0.4995 | <0.0001 |
| circTULP4 | - 0.6769 | 0.4582 | <0.0001 |
| circSLC8A1 | - 0.6722 | 0.4518 | <0.0001 |
| circVCAN | - 0.6710 | 0.4502 | <0.0001 |
| circZC3H6 | - 0.6474 | 0.4191 | <0.0001 |
| circCCDC9 | - 0.6280 | 0.3944 | <0.0001 |
| circPARD3 | - 0.5612 | 0.3149 | <0.0001 |
| circZNF91 | - 0.4877 | 0.2379 | <0.0001 |

Only significantly deregulated circRNAs with a fold change higher than 2 in lesional relative to non-lesional skin before treatment and normalization of expression upon treatment at day 84 are shown.

the circRNAs were down- or upregulated in lesional relative to non-lesional skin before treatment.

Of the 40 upregulated and six downregulated miRNAs with a fold change greater than two in lesional relative to non-lesional skin before treatment, ten showed a significant positive and four a significant negative correlation with the PASI, respectively (Table 2). For all considered miRNAs, we found the strongest correlation with the PASI for miR-223-3p and miR-4454 +miR-7975 with a correlation coefficient of 0.52 for both (Table 2, Figs 4C and S4B). In addition, both miR-203a-3p and miR-15a-5p, which were differentially expressed at day 42 and 84 in our global analysis, displayed a positive correlation with the PASI (Table 2, Figs 4D and S4B, respectively). However, none of these miRNAs showed a significant absolute expression change before 14 days of treatment (Figs 4D and S4B). Interestingly, a negative correlation was found between the significantly downregulated miRNAs in lesional skin at baseline and PASI changes, however, less pronounced (Table 2).

Like the findings for circRNAs, miRNA expression changes showed an either negative or positive correlation with the PASI during secukinumab treatment based on their initial direction of deregulation in lesional relative to non-lesional skin before treatment albeit less strong.

### miR-203a-3p expression changes correlate with total number of seed sites on the most abundant circRNAs in psoriasis during secukinumab treatment

Because miRNAs and circRNAs can interact [21,22,27] and were partially found to exhibit opposing expression patterns in psoriatic lesional compared to non-lesional skin and during secukinumab treatment, we investigated potential correlations between miRNA and circRNA expression patterns. However, most of the circRNAs harbour a low number of miRNA binding sites (BSs) [16], which would likely result in rather minor overall physiological consequences considering the relatively high abundance of miRNA target genes [25,51,52]. Nevertheless, circRNAs may also act in a cooperative manner to sequester miRNAs, assuming that several circRNAs within a subcellular structure contain seed sites for the same miRNAs [25,51]. Along this line, we predicted in a previous publication [16] miRNA seed sites on the circRNAs and found one miRNA to display a high number of cumulative BSs, namely miR-203a-3p. In the present study, we used circInteractome [53] for miRNA BS prediction. Here, we estimated the cumulative number of BSs for each considered miRNA using the sum of the product of the respective circRNA expression levels at the time and number of predicted BSs (Fig 5A). In this analysis, miR-203a and miR-626 stood out as being deregulated in lesional skin and having enriched numbers of BSs on the circRNAs. Specifically, we found 8 BSs for miR-203a and 12 for miR-626 (S8 File).

To explore a possible correlation between miR-203a/miR-626 and the cumulative number of BSs in our circRNAs during treatment, we used the same estimation for the cumulative number of miR-203a/miR-626 BSs as described above. When assessing the correlation between the average miR-203a-3p expression across the 14 patients relative to the log2-transformed values of the cumulative number of miR-203a BSs, we found a striking negative correlation over the course of the treatment (Fig 5B). This could also be confirmed when considering patient-specific expression values (Fig 5C and 5D and Table 3). In contrast, for miR-626 we observed a positive correlation (Fig 5E), albeit not significant, also when considering absolute expression values (Fig 5F).

Taken together, we found a strong negative correlation between the number of cumulative miR-203a BSs on the circRNAs and the absolute expression levels of miR-203a. On the other hand, miR-626 showed a comparable number of cumulative BSs but displayed a positive

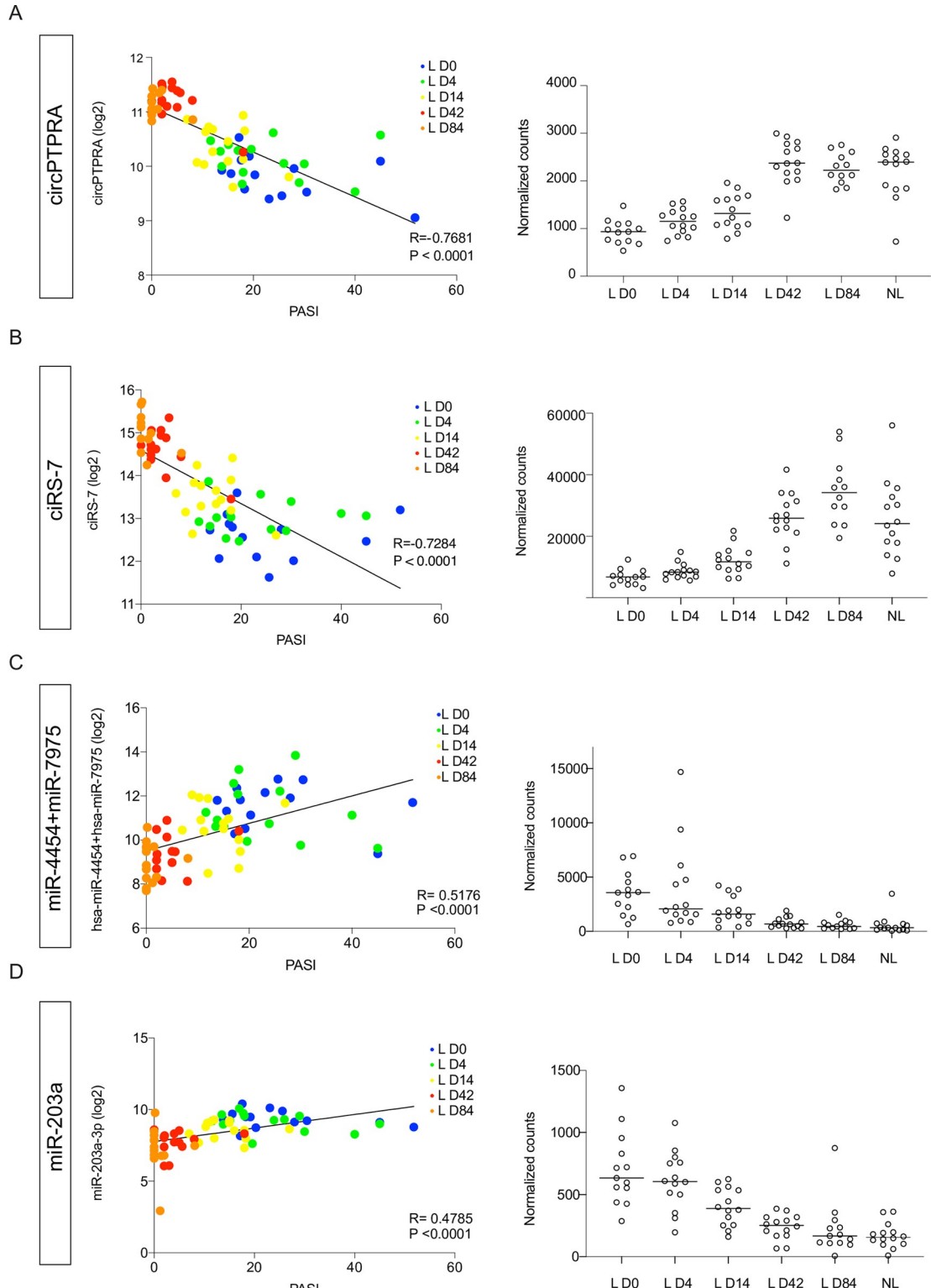

**Fig 4. Correlation between PASI and circRNA/miRNA expression in non-lesional (NL) and lesional (L) skin during secukinumab treatment.** (A-D; left) CircPTPRA (A), ciRS-7 (B), miR-4454+miR-7975 (C), and miR-203a-3p (D) log2-transformed expression values plotted against PASI during 84 days of secukinumab treatment. Simple linear regression was used for correlation between log2-transformed normalized counts and PASI. (A-D; right) CircPTPRA (A), ciRS-7 (B), miR-4454 +miR-7975 (C), and miR-203a-3p (D) expression in lesional skin during secukinumab treatment and non-lesional skin. Depicted

are normalized counts and median expression; n(NL, L D4, L D14, and L D43) = 14, n(L D0) = 13, n(circRNA-L D84) = 12, and n (miRNA-L D84) = 13.

correlation. Thus, if these correlations represent an active regulation of miRNAs through circRNA abundance, the circRNAs would need to protect miR-626 while causing degradation of miR-203a, indicating that these correlations are unlikely to imply causation.

## Discussion

Psoriasis is a multifaceted, chronic inflammatory skin disorder characterized by a complex pathogenesis, in which ncRNA classes, such as long ncRNAs and especially miRNAs, have been established to be deregulated and to play essential roles [15,48,54–56]. In addition, we have previously shown that global circRNA abundance is decreased in lesional relative to non-lesional skin of psoriatic patients [16,46]. However, no knowledge exists about miRNAs and circRNA expression dynamics upon anti-IL-17A treatment.

Here, we identified that secukinumab therapy over 84 days leads to a progressive normalization of all studied circRNAs that were deregulated at baseline. Intriguingly, already after the first secukinumab dose and only four days after treatment started, we observed a significant reversion of circRNA downregulation for the majority of considered transcripts. This predates clinical and histological improvements, which mainly occurred between day 14 and 42 as shown in our previous study using the same patient cohort [13]. In this previous study by Bertelsen *et al.*, we found 80 linear transcripts that were reversed already at day four of treatment [13]. In contrast to the present study these changes were greater than 2-fold. Nevertheless, the

**Table 2. Correlation between miRNA expression changes and PASI during 84 days of secukinumab treatment.**

| miRNA | R | R squared | p-value |
|---|---|---|---|
| miR-223-3p | 0.5241 | 0.2747 | <0.0001 |
| miR-4454+ miR-7975 | 0.5176 | 0.2679 | <0.0001 |
| miR-106a-5p+ miR-17-5p | 0.2833 | 0.2691 | <0.0001 |
|  |  |  |  |
| miR-15a-5p | 0.4828 | 0.2331 | <0.0001 |
| miR-203a-3p | 0.4785 | 0.2290 | <0.0001 |
| miR-132-3p | 0.3685 | 0.1358 | 0.0020 |
| miR-130a-3p | 0.3568 | 0.1273 | 0.0028 |
| miR-425-5p | 0.3541 | 0.1254 | 0.0031 |
| miR-200b-3p | 0.3538 | 0.1252 | 0.0031 |
| miR-93-5p | 0.3038 | 0.0923 | 0.0118 |
| miR-221-3p | 0.9022 | 0.0814 | 0.0184 |
| miR-22-3p | 0.2202 | 0.0485 | 0.0711 |
| miR-324-5p | 0.1970 | 0.0388 | 0.1073 |
| miR-146a-5p | 0.1819 | 0.0331 | 0.1378 |
| miR-455-3p | 0.1446 | 0.0209 | 0.2390 |
| miR-764 | - 0.3561 | 0.1628 | 0.0006 |
| miR-301b-3p | - 0.4681 | 0.2191 | <0.0001 |
| miR-412-3p | - 0.4618 | 0.2133 | <0.0001 |
| miR-144-3p | - 0.4570 | 0.2112 | <0.0001 |

Only significantly deregulated miRNAs with a fold change higher than 2 in lesional relative to non-lesional skin before treatment and normalization of expression upon treatment at day 84 are shown.

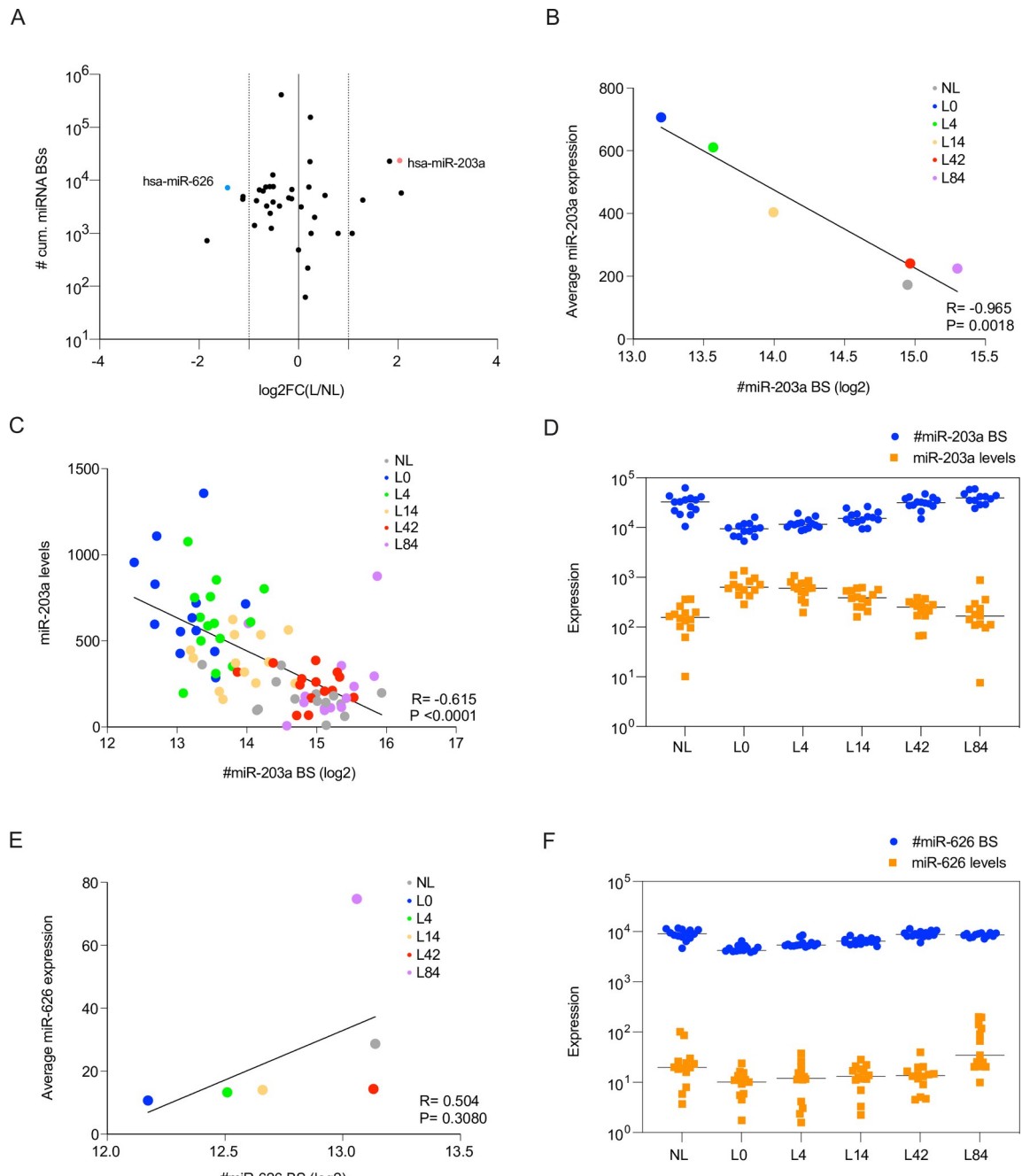

**Fig 5. MiR-203a-3p expression changes correlate with total number of seed sites on most abundant circRNAs in psoriasis during secukinumab treatment.** (A) Estimated number of total miRNA binding sites (BSs) for the most abundant circRNAs in psoriasis (on log10 scale) relative to the log2FC between lesional and non-lesional skin. The number of miRNA BSs was estimated by the sum over the product of all circRNAs with predicted unique miRNA BSs and their respective average circRNA expression level at the time point. (B-C) Correlation of average (B) or patient-specific (C) miR-203a-3p expression over the course of treatment with a proxy for the total number of miR-203a BSs present in the most abundant circRNAs in psoriasis (see description for estimation in A). Simple linear regression was used for correlation between average miR-203a-3p expression (y-axis) and the log2-transformed values of the predicted number of miR-203a BSs across all circRNAs (x-axis). (D) Opposing absolute changes (on log10 scale) in miR-203a-3p expression levels (yellow) and estimated number of circRNA-specific miR-203a seed sites during secukinumab treatment. (E) Correlation of average miR-626 expression over the course of treatment with a proxy for the total number of miR-626 BSs present in the most abundant circRNAs in psoriasis (see description for estimation in A). Simple linear regression was used for establishing the correlation between average miR-626 expression (y-axis) and the log2-transformed values of the predicted number of miR-626 BSs across all circRNAs (x-axis). (F) Matching absolute expression changes (on log10 scale) in miR-626 expression levels (yellow) and estimated number of circRNA-specific miR-626 seed sites during secukinumab treatment.

**Table 3. Correlation between miR-203a-3p expression and cumulative miR-203a BSs on the circRNAs in individual patients during secukinumab treatment.**

| Patient | R | R squared | p-value | Deviation from zero? |
|---|---|---|---|---|
| 1 | 0.9841 | 0.9684 | 0.0024 | significant |
| 2 | 0.9308 | 0.8664 | 0.0070 | significant |
| 3 | 0.4862 | 0.2364 | 0.4063 | ns |
| 4 | 0.7336 | 0.5381 | 0.0970 | ns |
| 5 | 0.9038 | 0.8169 | 0.0134 | significant |
| 6 | 0.7612 | 0.5794 | 0.0788 | ns |
| 7 | 0.8555 | 0.7318 | 0.0298 | significant |
| 8 | 0.9864 | 0.9730 | 0.0003 | significant |
| 9 | 0.8410 | 0.7072 | 0.0359 | significant |
| 10 | 0.8002 | 0.6403 | 0.0559 | ns |
| 11 | 0.9426 | 0.8885 | 0.0048 | significant |
| 12 | 0.6151 | 0.3783 | 0.1937 | ns |
| 13 | 0.8744 | 0.7646 | 0.0227 | significant |
| 14 | 0.9135 | 0.8345 | 0.0301 | significant |

Values of the estimated numbers of miR-203a BSs on the circRNAs were log2-transformed. Ns = not significant.

early upregulation of all circRNAs is noteworthy and opposed to our observations of miRNA expression changes in the same skin biopsies and time points, where we observed a less pronounced and later reversal of individual miRNAs. A potential explanation for these discrepancies could be that a circRNA-specific regulation through biogenesis and/or decay occurs, or that an active regulation of miRNA and mRNA transcripts arises. On one hand, previous publications demonstrated that global circRNA levels can be influenced by viral infection triggering RNAse L-mediated degradation [37] or upon modulation of circRNA biogenesis through RBP interaction [20,45]. Although we observed no changes in the transcript levels of five mRNAs coding for circRNA biogenesis regulators, additional endonucleases [57,58] or RBPs [28,59] could have an impact on global expression levels. On the other hand, circRNA abundance was previously found to be higher in further differentiated cells compared to stem cell states [32,60]. This has prompted the hypothesis that circRNAs accumulate within post-mitotic cells due to their higher stability compared to mRNA, as well as a dilution of circRNAs within more proliferative cells [61]. In our previous study by Bertelsen *et al.* [13], we investigated the normalization of the proliferation marker Ki67 over the 84-day treatment course and found hyperproliferative keratinocytes indicated by high Ki67 levels in the basal layer of the epidermis in lesional skin at day 0, 4, and 14. From day 42 onward this hyperproliferation disappeared. Therefore, although no histological indication of a reduction in keratinocyte proliferation was observed by Ki67 staining at day four [13], minor changes in proliferation could already be reflected in a global upregulation of circRNA levels. Even if circRNA expression changes in psoriasis pathogenesis and during treatment are a sole result of dynamics in keratinocyte proliferation levels and thus dilution, they could still have molecular consequences for surrounding RNAs, RBPs, or ion distribution. In addition, with miRNAs and mRNAs being subject to exonucleolytic degradation pathways, these transcripts might be subject to a more active regulation and could therefore be less globally affected by the commenced treatment.

As opposed to the circRNA alterations in skin biopsies, we found no significant changes in circRNA levels in the patients PBMCs. Although specific miRNAs detected in PBMCs were suggested to harbor particular biomarker potential [50], our findings appeared in line with

previous findings on the same patient cohort, in which no alterations in the linear transcriptome was found in these cells [13].

The majority of circRNAs correlated stronger with the PASI than observed for the investigated miRNAs. Some of the most deregulated circRNAs during treatment included circPTPRA, circTULP4, and ciRS-7, all three of which also correlated negatively with the PASI and could thus be potentially useful in identifying the disease severity and post-treatment disease resolution. To this end, it is however a limiting factor that skin biopsies would be required, as no circRNA changes in PBMCs were detectable.

Moreover, the spatial ciRS-7 expression patterns showed a drastic signal loss in the epidermis of lesional skin before treatment with a simultaneous increase in ciRS-7 signal in the dermis and dermal papillae. As the treatment advanced, ciRS-7 levels reversed in both epidermis and dermis. Interestingly, in non-lesional skin before treatment, ciRS-7 was predominantly found in the basal layer of the epidermis, in which epidermal stem cells reside, and to a lesser extent in the spinous and granular layer. As mentioned above, circRNA levels are low in proliferative cells and could thus point towards a specific function for ciRS-7 in epidermal stem cells.

Although most of the circRNAs were significantly upregulated during treatment, two circRNAs became downregulated, namely circERC1 and circCAMSAP1. These circRNAs were slightly upregulated at baseline and it could be speculated that they might be produced at a faster rate and therefore not diluted during cell proliferation [62].

Additionally, we found several miRNAs, including miR-203a [47,48] and miR-223 [50], to be reversed in their expression levels relative to baseline during treatment. In particular, miR-203a-3p was shown to be enriched in psoriatic skin lesions and to possess a critical role in disease progression by targeting genes involved in epidermal differentiation, such as *p63* [63] and *SOCS-3* [15], and in immune responses with targets like pro-inflammatory cytokines TNFα and IL-24 encoding transcripts [47]. In a recent study [56], miR-378a ectopic overexpression, mimicking the miRNA increase in psoriasis, was found to promote p63 transport to the nucleus and further contributed to the IL-17A-mediated activation of NF-κB pathway signaling by acting as suppressor of *NFKBIA*. While miR-378a was not included in the miRNA NanoString panel used in this study, we found miR-378i to be deregulated in skin biopsies. To the best of our knowledge, miR-378i has not been linked to psoriatic lesions previously and does not share seed site similarities with miR-378a. In addition, we found miR-93-5p as an interesting miRNA candidate. It belongs to the miR-106b-25 cluster, which is a paralog of the polycistronic miR-17-92 cluster and located within an intronic region of *MCM7*, and shares seed sites with miR-17, miR-20a/b, and miR-106a/b [64]. Strikingly, although miR-93 has not been directly implicated in psoriasis pathogenesis, expression of the miR-17-92 cluster was found to be increased in psoriasis and to promote keratinocyte proliferation and chemokine production [65]. Also, miR-93 itself was linked to the regulation of keratinocyte migration, proliferation, and wound healing [49] and could thus possess regulatory capacities in psoriasis pathogenesis. However, future research is needed to delineate the regulatory networks these miRNAs play in the process.

Given that miR-203a is amongst the most extensively studied miRNAs in the context of psoriasis [15,47,48,63], we were intrigued to find one of the highest cumulative numbers of BSs for this miRNA on the circRNAs investigated here [16] as well as a marked correlation between the cumulative abundance of BSs on six circRNAs and the absolute expression of miR-203a. Such a cooperative model of action for several circRNAs [25,51] would be especially reasonable in light of the synchronized deregulation of circRNA levels during secukinumab treatment. However, in the present study we solely demonstrate a correlation between miR-203a levels and circRNA BS abundance, which could be a secondary effect of pathogenic

processes. Especially, when considering the rather positive correlation between a miRNA with similar cumulative BSs, miR-626, and circRNA BS abundance. Nevertheless, further combinatorial loss- or gain-of-function studies could shed light on potential cooperative effects of circRNAs on miRNA expression.

In summary, we provide knowledge about miRNA and circRNA dynamics during anti-IL-17A treatment in psoriasis. Despite the lack of histological improvements after four days of treatment, we found that circRNAs, but not miRNAs, were already globally upregulated in skin samples at this timepoint, demonstrating the rapid action of secukinumab and further underlining the biomarker potential of circRNAs. This was further underpinned by a strong correlation of ciRS-7, circPTPRA, and circTULP4 with the PASI.

## Supporting information

**S1 Fig. Alterations in global circRNA expression in non-lesional (NL) and lesional (L) skin of psoriasis patients before and during secukinumab treatment.** (A) Principal component analysis based on circRNA expression levels in paired lesional and non-lesional patients before treatment and healthy control skin (NN). (B) Volcano plot showing changes circRNA expression in non-lesional psoriasis skin before treatment relative to healthy control skin (NN). Plots depict adjusted (adj.) p-values relative to log2FC. Multiple unpaired t-test with correction for multiple comparisons (FDR-Benjamini-Hochberg). (C) Heatmap with unsupervised hierarchical clustering of circRNA expression (as z-score of log-transformed values) for individual patients from non-lesional and paired lesional psoriasis skin dependent on the day of treatment. (D) Volcano plot showing changes circRNA expression in lesional psoriasis skin before treatment relative to healthy control skin (NN). Plots depict adjusted (adj.) p-values relative to log2FC. Multiple unpaired t-test with correction for multiple comparisons (FDR-Benjamini-Hochberg). (E) Volcano plot showing changes in specific circRNA and mRNA expression in lesional psoriasis skin at day 84 of treatment (L D84) compared to non-lesional skin. Multiple paired t-test with correction for multiple comparisons (FDR-Benjamini-Hochberg). For panels C and E: n(NL, L D4, L D14, and L D43) = 14, n(L D0) = 13, and n(L D84) = 12; for panels A, B, and D: n = 8.
(PDF)

**S2 Fig. CircRNA expression in PBMCs of psoriasis patients before and after secukinumab treatment.** (A) Heatmap using unsupervised hierarchical clustering of circRNA expression levels (as z-score of log-transformed values) in peripheral blood mononuclear cells (PBMCs) from patients before (D0; n = 12) and after 4 (D4, n = 13), 14 (D14; n = 12), 42 (D42; n = 12), and 84 (D84; n = 13) days of treatment. (B) Principal component analysis based on circRNA expression levels in peripheral blood mononuclear cells (PBMCs) from patients before (D0) and after 4 (D4), 14 (D14), 42 (D42), and 84 (D84) days of treatment. (C-F) Volcano plot showing changes in circRNA expression in PBMCs from patients at day four (D4; C), day 14 (D14; D), (D42; E), and day 84 (D84; F) in contrast to day 0 (D0) of secukinumab treatment. Depicted are p-values relative to the log2FC. Multiple paired t-test without correction for multiple comparisons. (G-H) CiRS-7 (G) and circPTPRA (H) expression in PBMCs during 84 days of secukinumab treatment. Depicted are normalized counts and median expression. One-way ANOVA with correction for multiple comparisons (FDR-Benjamini-Hochberg); n (D0, D14, D42) = 12 and n(D4, D84) = 13.
(PDF)

**S3 Fig. Changes in miRNA expression profile in psoriatic skin during secukinumab treatment.** (A) Principal component analysis based on mean miRNA expression levels between

patients from non-lesional and paired lesional psoriasis skin before (L D0) and after 4 (L D4), 14 (L D14), 42 (L D42), and 84 (L D84) days of treatment. (B) Principal component analysis based on miRNA expression levels in paired lesional and non-lesional patients before treatment and healthy control skin (NN). (C) Volcano plot showing changes miRNA expression in non-lesional psoriasis skin before treatment relative to healthy control skin (NN). Plots depict adjusted (adj.) p-values relative to log2FC. Multiple unpaired t-test with correction for multiple comparisons (FDR-Benjamini-Hochberg). (D-E) Volcano plot showing changes in miRNA expression in day 42 (D) and 84 samples (E) in contrast to non-lesional skin. Multiple paired t-test with correction for multiple comparisons (FDR-Benjamini-Hochberg). (F) Volcano plot showing changes miRNA expression in lesional psoriasis skin before treatment relative to healthy control skin (NN). Plots depict adjusted (adj.) p-values relative to log2FC. Multiple unpaired t-test with correction for multiple comparisons (FDR-Benjamini-Hochberg). For panels A, D, and E: n(NL, L D4, L D14, and L D43) = 14, n(L D0 and L D84) = 13; for panels B, C, and F: n(NL, L) = 8 and n(NN) = 7. Depicted are miRNAs with an average expression above 20 counts (n = 58).
(PDF)

**S4 Fig. Correlation between PASI and circPTPRA skin expression in non-lesional (NL) and lesional (L) skin during secukinumab treatment.** (A+B; left) CircTULP4 (A), miR-223-3p (B), and miR-15a-5p (C) log2-transformed expression values plotted against PASI during 84 days of secukinumab treatment. Simple linear regression was used for correlation between log2-transformed normalized counts and PASI. (A+B; right) CircTULP4 (A), miR-223-3p (B), and miR-15a-5p (C) expression in lesional skin during secukinumab treatment and non-lesional skin. Depicted are normalized counts and median expression; n(NL, L D4, L D14, and L D43) = 14, n(L D0) = 13, n(circRNA-L D84) = 12, and n(miRNA-L D84) = 13.
(PDF)

**S1 File. List of targets and probes for custom-designed circRNA NanoString panel.**
(XLSX)

**S2 File. Raw and normalized circRNA data for non-lesional and lesional skin of psoriasis patients before and during secukinumab treatment.**
(XLSX)

**S3 File. Raw and normalized circRNA data for paired non-lesional and lesional skin of psoriasis patients before secukinumab treatment and healthy control skin.**
(XLSX)

**S4 File. Raw and normalized circRNA data for PBMCs isolated from psoriasis patients before and during secukinumab treatment.**
(XLSX)

**S5 File. Raw and normalized miRNA data for non-lesional and lesional skin of psoriasis patients before and during secukinumab treatment.**
(XLSX)

**S6 File. Raw and normalized miRNA data for paired non-lesional and lesional skin of psoriasis patients before secukinumab treatment and healthy control skin.**
(XLSX)

**S7 File. List of significantly upregulated circRNAs at baseline and corresponding PASI correlation.**
(XLSX)

**S8 File. Estimation of cumulative number of BSs for each considered miRNA using the sum of the product of the respective circRNA expression levels at the time and number of predicted BSs based on circInteractome BS prediction.**
(XLSX)

## Author Contributions

**Conceptualization:** Sabine Seeler, Liviu-Ionut Moldovan, Trine Bertelsen, Lars Iversen, Claus Johansen, Jørgen Kjems, Lasse Sommer Kristensen.

**Formal analysis:** Sabine Seeler, Liviu-Ionut Moldovan.

**Funding acquisition:** Jørgen Kjems, Lasse Sommer Kristensen.

**Investigation:** Sabine Seeler, Liviu-Ionut Moldovan, Trine Bertelsen, Henrik Hager, Claus Johansen.

**Project administration:** Lasse Sommer Kristensen.

**Resources:** Henrik Hager, Lars Iversen, Claus Johansen, Jørgen Kjems, Lasse Sommer Kristensen.

**Supervision:** Jørgen Kjems, Lasse Sommer Kristensen.

**Visualization:** Sabine Seeler, Lasse Sommer Kristensen.

**Writing – original draft:** Sabine Seeler, Liviu-Ionut Moldovan.

**Writing – review & editing:** Trine Bertelsen, Henrik Hager, Lars Iversen, Claus Johansen, Jørgen Kjems, Lasse Sommer Kristensen.

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
