## [Decision Letter · Decision Letter 0]

3 Aug 2022

PONE-D-22-15625Global circRNA expression changes predate clinical and histological improvements of psoriasis patients upon secukinumab treatmentPLOS ONE

Dear Dr. Sommer Kristensen,

Thank you for submitting your manuscript to PLOS ONE. After careful consideration, we feel that it has merit but does not fully meet PLOS ONE’s publication criteria as it currently stands. Therefore, we invite you to submit a revised version of the manuscript that addresses the points raised during the review process.

We look forward to receiving your revised manuscript.

Kind regards,

Irina Budunova

Academic Editor

PLOS ONE

Journal Requirements:

“I have read the journal's policy and the authors of this manuscript have the following competing interests: L.I. served as a consultant and/or paid speaker for and/or participated in clinical trials sponsored by AbbVie, Almirall, Amgen, AstraZeneca, BMS, Boehringer Ingelheim, Celgene, Centocor, Eli Lilly, Janssen Cilag, Kyowa, Leo Pharma, MSD, Novartis, Pfizer, Samsung, and UCB.”

Additional Editor Comments:

The authors characterized the expression of non-coding RNAs: circular RNAs and miRNAs in psoriasis lesional skin during the chronic 84 day treatment with Secukinumab ( IL17 Ab). They found that Secukinumab did not significantly alter circRNA expression patterns in PBMCs, but did efficiently reduced their expression in psoriatic lesional skin. They also reported here that circRNA expression normalization in lesional skin during the treatment occurred faster than normalization of mRNA expression. As the expression changes of non-coding RNAs and especially circRNAs during psoriasis treatments are not well known , this paper is timely and important. The finding that circRNA expression is stabilized during Secukinumab treatment course before mRNA changes is novel. The bioinformatics analysis of mRNA and circRNA expression and correlation with PASI is sound. However, there are several deficiencies in data presentation and the scope of the study that require major revision.

It is important to compare the level of “normalized” non-coding RNA expression after the Secukinumab not only with non-lesional but also with normal skin. It is also interesting whether and how non-coding RNA expression changes in non-lesional skin.

In Fig. 1C, the circRNA should be called different from the gene name, not HIPK3 but circHIPK3 for example.

Fig. 2 A. Red color (miRNA) is very difficult to see.

Fig. 2 B legend is incorrect: “The visualization of Hematoxylin and Eosin staining combined with RNA chromogenic in situ hybridization (CISH)”. The authors did not use eosin , only liquid

permanent red (DAKO, Glostrup, Denmark) and Mayer’s hematoxylin counterstaining.

Fig. 2B. The differences in ciRS-7 are not very convincing: LD84 does not look that different from LD84 The authors should do morphometric analysis of CISH signal in the lesional skin .

Reviewers' comments:

Reviewer's Responses to Questions

**Comments to the Author**

1. Is the manuscript technically sound, and do the data support the conclusions?

Reviewer #1: Yes

Reviewer #2: Yes

2. Has the statistical analysis been performed appropriately and rigorously? 

Reviewer #1: Yes

Reviewer #2: Yes

3. Have the authors made all data underlying the findings in their manuscript fully available?

Reviewer #1: Yes

Reviewer #2: Yes

4. Is the manuscript presented in an intelligible fashion and written in standard English?

Reviewer #1: Yes

Reviewer #2: Yes

5. Review Comments to the Author

Reviewer #1: This article characterizes a set of circRNAs, miRNAs, and a few mRNAs in normal vs psoriatic skin in patients undergoing ixekizumab treatment, which is highly effective against psoriasis. The experiments are technically sound. As might be expected, abnormally-expressed circRNA and miRNA expression levels return towards those seen in pretreatment non-lesional (NL) skin. The finding that the circlRNAs normalize more rapidly than their mRNAs is interesting. Other than this, there are no dramatic specific findings that emerge from this study. As there are known abnormalities in NL skin compared to normal control (NN) skin, the paper would be strengthened by a comparison of NL skin to NN skin.

Reviewer #2: The original research work of Kristensen’s Laboratory “Global circRNA expression changes predate clinical and histological improvements of psoriasis patients upon secukinumab treatment” has shown the involvement of non-coding RNA and, in particular circular RNA, in psoriasis pathogenesis and a possibility to normalize the level of a set of circRNA in lesional skin by secukinumab, the first targeted anti-IL-17A drag for psoriasis treatment. These priority findings open opportunities for the use of circRNAs as predictive markers for treatment efficacy and as prognostic markers for disease progression. Moreover, it reveals novel approaches for the treatment of psoriasis directed to normalizing of ncRNA imbalances. Design of the study is well done and the methods used are perfectly corresponded to the aims. All the protocols for experimental procedures are provided in details. Modern statistical analysis has been applied for the data processing. The results have been described and discussed properly and well presented in figures and tables. Some data of the study, especially concerning micro RNA dynamics during psoriasis progression, are in accordance with the previously published one, and the whole concept of the study is in agreement with current knowledge of circular RNAs. Thus, this work may be recommended for acceptance for publication in PLOS One journal as the original research article. However, some minor revision of the text is required:

1. Authors should check whether all the abbreviations are included in the text. As an example, there is no abbreviation for peripheral blood mononuclear cells – PBMC.

2. There is correct description of the histologic slide preparing, however there is a mistake in the Figure 2 legend (hematoxylin-eosin staining is pointed out).

3. There is a typing error on the page 7, line 1: it should be “All p-values” instead of “All P-values”.

4. The authors may discuss correlation of their data with the results of Bertelsen et al., 2020 concerning visible Ki67 staining up to 14 day of the treatment with secukinumab, but disappearance of Ki67 staining since 42 day of secukinumab treatment (ref. 13).

5. Pairing of the groups provided in the figures should be done more carefully as it should correspond to the logic of the study, for example in the figure 1 the columns NL and L D84 should change places. I would recommend to check all the figures relatively to this point.

6. PLOS authors have the option to publish the peer review history of their article (what does this mean?). If published, this will include your full peer review and any attached files.

Reviewer #1: No

Reviewer #2: **Yes: **Marianna G. Yakubovskaya

---

## [Author Response · Author response to Decision Letter 0]

8 Sep 2022

Author’s response to reviewers

Dear Editors,

We thank you and the reviewers for the helpful comments and suggestions on our manuscript entitled “Global circRNA expression changes predate clinical and histological improvements of psoriasis patients upon secukinumab treatment”. 

Please find hereafter the detailed response to all comments as well as attached the revised manuscript with and without track changes. We have addressed the raised concerns and edited the manuscript accordingly. 

We believe that the incorporated changes improved the manuscript substantially and hope you will find the revised version satisfactory as well.

Reviewer #1: 

This article characterizes a set of circRNAs, miRNAs, and a few mRNAs in normal vs psoriatic skin in patients undergoing ixekizumab treatment, which is highly effective against psoriasis. The experiments are technically sound. As might be expected, abnormally-expressed circRNA and miRNA expression levels return towards those seen in pretreatment non-lesional (NL) skin. The finding that the circlRNAs normalize more rapidly than their mRNAs is interesting. Other than this, there are no dramatic specific findings that emerge from this study. As there are known abnormalities in NL skin compared to normal control (NN) skin, the paper would be strengthened by a comparison of NL skin to NN skin.

Author’s response: 

We thank the reviewer for the suggestion and agree that comparing non-coding RNA (ncRNA) expression between NL and NN skin could benefit the understanding of miRNA and circRNA dynamics in the skin of psoriasis patients. Therefore, we performed an additional characterization of circRNA and miRNA patterns in eight healthy control (NN) as well as eight paired non-lesional (NL) and lesional (L) skin biopsies using NanoString nCounter technology. To this end, we used skin samples of 8 out of the 14 patient cohort and compared the expression levels to eight age- and gender- matched NN skin samples. Here, we found no significant differences in either circRNA (panels A-B in S1 Fig) or miRNA profile (panels B-C in S3 Fig) between NL and NN samples. Moreover, we included a comparison of circRNA changes in L skin before treatment to NN (panel D in S1 Fig) and found amongst others ciRS-7 and circPTPRA to be decreased, as seen for L D0 relative to NL skin (Fig 2A; left). For miRNAs, we also included a volcano plot showing miRNAs that are upregulated in L D0 skin compared to NN (panel F in S3 Fig), including miR-203a, miR-378i, and miR-4454+miR-7975, as seen for L D0 relative to NL (Fig 3F; left). 

We refer to the added figure panels on page 8-line3/16 (“Thus, we also investigated if circRNA patterns showed variation between non-lesional and healthy control skin, but did not find significant differences (S1A-B Fig, S3 File).” and “[…]as well as relative to healthy control skin (S1D Fig).”) and page 12-line 9/15 (“In addition, we again checked for miRNA changes between non-lesional and healthy control skin, but found no significant differences (S3B-C Fig, S6 File).” and “[…], but healthy control skin (S3F Fig, S6 File).”)

For detailed description of patient and control cohort, please see section “Patient cohort and sample preparation” (page 4-line 18). Here we included the sentence “In addition, skin biopsies of eight healthy age- and gender-matched (average age = 38.38 years; n[male controls] = 5, n[female controls] = 3) controls were compared to non-lesional/lesional samples from 8 out of the 14 psoriasis patients (average age = 42.38 years; n[male controls]= 6, n[female controls] = 2).”. For NanoString experiments, we would like to refer to sections “MiRNA expression analysis using NanoString nCounter technology” (page 6) and “CircRNA expression analyses using NanoString nCounter technology” (page 5). We upated the section “Statistical analysis” (page 7-line 5) with the sentence “For comparison of lesional or non-lesional to healthy control skin, we performed multiple unpaired t-tests with correction for multiple comparisons (FDR-Benjamini-Hochberg).” We attached two additional supplementary information files in accordance with these additional experiments, one for the circRNA data (S3 File) and one for the miRNA data (S6 File). 

In addition, we updated the list of NanoString targets and probes (S1 File), because an updated NanoString panel including seven additional circRNA targets was used for the experiment. To avoid batch effects, we included all 24 samples in this additional experiment. 

Marianna G. Yakubovskaya (Reviewer #2): 

The original research work of Kristensen’s Laboratory “Global circRNA expression changes predate clinical and histological improvements of psoriasis patients upon secukinumab treatment” has shown the involvement of non-coding RNA and, in particular circular RNA, in psoriasis pathogenesis and a possibility to normalize the level of a set of circRNA in lesional skin by secukinumab, the first targeted anti-IL-17A drag for psoriasis treatment. These priority findings open opportunities for the use of circRNAs as predictive markers for treatment efficacy and as prognostic markers for disease progression. Moreover, it reveals novel approaches for the treatment of psoriasis directed to normalizing of ncRNA imbalances. Design of the study is well done and the methods used are perfectly corresponded to the aims. All the protocols for experimental procedures are provided in details. Modern statistical analysis has been applied for the data processing. The results have been described and discussed properly and well presented in figures and tables. Some data of the study, especially concerning micro RNA dynamics during psoriasis progression, are in accordance with the previously published one, and the whole concept of the study is in agreement with current knowledge of circular RNAs. Thus, this work may be recommended for acceptance for publication in PLOS One journal as the original research article. However, some minor revision of the text is required:

1. Authors should check whether all the abbreviations are included in the text. As an example, there is no abbreviation for peripheral blood mononuclear cells – PBMC.

Author’s response: 

We would like to thank the reviewer for carefully evaluating the manuscript. We have now checked if all abbreviations are specified when first mentioned. The abbreviations for peripheral blood mononuclear cells (PBMCs) and backsplicing junction (BSJ) are now included on page 4- line 9 and on page 5- line 21 respectively. 

2. There is correct description of the histologic slide preparing, however there is a mistake in the Figure 2 legend (hematoxylin-eosin staining is pointed out).

Author’s response: 

We thank the reviewer for pointing out this mistake. The corrected figure legend can be found on page 10- line 3. 

3. There is a typing error on the page 7, line 1: it should be “All p-values” instead of “All P-values”.

Author’s response: 

Thank you to the reviewer for pointing out this error. We corrected it accordingly to “All p-values were two-tailed and considered significant if p < 0.05.” (page 7-line 9)

4. The authors may discuss correlation of their data with the results of Bertelsen et al., 2020 concerning visible Ki67 staining up to 14 day of the treatment with secukinumab, but disappearance of Ki67 staining since 42 day of secukinumab treatment (ref. 13).

Author’s response: 

We agree with the reviewer that the discussion regarding a potential circRNA dilution based on keratinocyte hyperproliferation could be explained more comprehensive. Therefore, we included the following paragraph on page 20: 

“In our previous study by Bertelsen et al. […], we investigated the normalization of the proliferation marker Ki67 over the 84-day treatment course and found hyperproliferative keratinocytes indicated by high Ki67 levels in the basal layer of the epidermis in lesional skin at days 0, 4, and 14. From day 42 onward this hyperproliferation disappeared. Therefore, although no histological indication of a reduction in keratinocyte proliferation was observed by Ki67 staining at day four […], minor changes in proliferation could already be reflected in a global upregulation of circRNA levels.” 

5. Pairing of the groups provided in the figures should be done more carefully as it should correspond to the logic of the study, for example in the figure 1 the columns NL and L D84 should change places. I would recommend to check all the figures relatively to this point.

Author’s response: 

Overal, we agree with the reviewer that sample groups in figure panels should be organized in a logical manner. However, all heatmaps within this manuscript (Fig 1C, 3B, S1C, and S2A) are illustrating an unsupervised hierarchical clustering approach to determine sample clusters with similar circRNA or miRNA expression patterns in an unbiased way. Hence, re-organizing the columns would contradict this unsupervised approach. 

Irina Budunova (Academic Editor):

The authors characterized the expression of non-coding RNAs: circular RNAs and miRNAs in psoriasis lesional skin during the chronic 84 day treatment with Secukinumab ( IL17 Ab). They found that Secukinumab did not significantly alter circRNA expression patterns in PBMCs, but did efficiently reduced their expression in psoriatic lesional skin. They also reported here that circRNA expression normalization in lesional skin during the treatment occurred faster than normalization of mRNA expression. As the expression changes of non-coding RNAs and especially circRNAs during psoriasis treatments are not well known , this paper is timely and important. The finding that circRNA expression is stabilized during Secukinumab treatment course before mRNA changes is novel. The bioinformatics analysis of mRNA and circRNA expression and correlation with PASI is sound. However, there are several deficiencies in data presentation and the scope of the study that require major revision.

It is important to compare the level of “normalized” non-coding RNA expression after the Secukinumab not only with non-lesional but also with normal skin. It is also interesting whether and how non-coding RNA expression changes in non-lesional skin.

Author’s response: 

We thank the academic editor for the helpful suggestion. As described above in the response to the first reviewer’s comment, we carried out additional experiments to characterize the circRNA and miRNA patterns between NL and NN skin. We decided, however, against the normalization of all time-points to NN skin, because the NanoString experiments were conducted with different codeset batches, which would potentially introduce a significant bias based on batch effects instead of showing true physiological differences. Moreover, based on the comparison of NN to NL and L skin before treatment commenced, we already see a high similarity in circRNA as well as miRNA expression between NN and NL skin, which speaks for the validity of the findings seen during the course of secukinumab treatment between L and NL skin. 

In Fig. 1C, the circRNA should be called different from the gene name, not HIPK3 but circHIPK3 for example.

Author’s response: 

We would like to thank the editor for pointing out this inconsistency. We changed the naming of circRNAs in all heatmaps within the manuscript (Fig 1C, S1C, S2A, and S1E). 

Fig. 2 A. Red color (miRNA) is very difficult to see.

Author’s response: 

We agree with the editor’s comment and changed the color of mRNAs in all plots accordingly (Fig 1D-F, and 2A). We hope that this change improved the data point visibility. 

Fig. 2 B legend is incorrect: “The visualization of Hematoxylin and Eosin staining combined with RNA chromogenic in situ hybridization (CISH)”. The authors did not use eosin , only liquid

permanent red (DAKO, Glostrup, Denmark) and Mayer’s hematoxylin counterstaining.

Author’s response: 

We thank the editor for pointing out this mistake. As mentioned above, the corrected figure legend is included on page 10- line 3.

Fig. 2B. The differences in ciRS-7 are not very convincing: LD84 does not look that different from LD84 The authors should do morphometric analysis of CISH signal in the lesional skin .

Author’s response:

We thank the editor for this suggestion. Firstly, we would like to mention that we were not sure, to which time point the editor is refering to when comparing the ciRS-7 expression to L D84. Therefore, we responded to this comment rather broadly. 

While we agree that a quantitiative analysis of ciRS-7 signal in the epidermis and dermis in NL and L skin during secukinumab treatment would be benefical to this study, we believe that the analysis result for this CISH experiment would not be robust or conclusive due to technical difficulties. For a robust quantification, a defined particle size and clear separation of overlapping signals would be needed. In our CISH experiments, especially within the dermis, we cannot clearly distinguish single ciRS-7 molecules, which makes a quantification difficult. In addition, we believe that the conclusions we are drawing from the CISH experiment are detectable without additional morphometric analysis, as we state in the manuscript: “While we observed ciRS-7 in non-lesional skin to be predominately expressed in the epidermis, especially within the basal layer, a marked downregulation of ciRS-7 in the epidermis of lesional skin was found (Fig 2B), in line with previous observations […]. On the contrary, we found a strong ciRS-7 presence in the dermal compartments of lesional skin, particularly before and early during treatment. Thus, the observed ciRS-7 spatial expression patterns during treatment confirmed a progressive restoration of ciRS-7 levels in the epidermis over 84 days of treatment, further supporting our NanoString experiments.”. We hope this explains our rational on why we decided against a quantification analysis of ciRS-7 levels. 

Additional author comments:

Following PLOS ONE guidelines, we also updated the ethical statement regarding patient consent in the materials and methods section to the sentences “The study was carried out following the Declaration of Helsinki and a written and signed informed consent was obtained from each patient. Only adult patients above the age of 18 were enrolled. The study was approved by The Central Denmark Region Committees on Health Research Ethics (M-20090102).” (page 4-5). 

In addition, we have thoroughly checked and edited for mistakes in spelling, grammar, and in-text citations to tables, files, and figures, as well as errors in the reference list. 

Also, we realized that we did not include the raw and normalized data on circRNA expression in PBMCs in the first submission. We would like to apologize for this mistake and have included the data now in this revised manuscript submission (S4 File). 

We included the supporting information captions to the end of the manuscript according to PLOS ONE guidelines. The figure legends were updated according to the added panels to the supplementary figures. 

Lastly, we modiefied the manuscript according to the PLOS ONE style requirements. These included formatting changes, specifically changes in font size, table style, positioning of figure legends, formatting of title page, and reference style. However, these changes are for simplicity reasons not indicated with Track changes in the document “Revised manuscript with track changes”.

---

## [Editor Report · Decision Letter 1]

13 Sep 2022

Global circRNA expression changes predate clinical and histological improvements of psoriasis patients upon secukinumab treatment

PONE-D-22-15625R1

Dear Dr. Kristensen,

We’re pleased to inform you that your manuscript has been judged scientifically suitable for publication and will be formally accepted for publication once it meets all outstanding technical requirements.

Kind regards,

Irina Budunova

Academic Editor

PLOS ONE

Additional Editor Comments (optional):

Thank you for submitting your manuscript to the PlosOne. We are pleased that you considered us for publication of your work.

In the judgment of the editors, the manuscript was substantially revised, all reviews’ comments have been addressed. The revised manuscript is acceptable for publication.
---

## [Editor Report · Acceptance letter]

20 Sep 2022

PONE-D-22-15625R1 

Global circRNA expression changes predate clinical and histological improvements of psoriasis patients upon secukinumab treatment

Dear Dr. Sommer Kristensen:

I'm pleased to inform you that your manuscript has been deemed suitable for publication in PLOS ONE. Congratulations! Your manuscript is now with our production department. 

Kind regards, 

on behalf of

Dr. Irina Budunova 

Academic Editor

PLOS ONE